# Global economic costs due to vivax malaria and the potential impact of its radical cure: A modelling study

**Angela Devine**[1,2]*, **Katherine E. Battle**[3], **Niamh Meagher**[2,4], **Rosalind E. Howes**[5,6], **Saber Dini**[2], **Peter W. Gething**[7,8], **Julie A. Simpson**[2], **Ric N. Price**[1,9,10◦], **Yoel Lubell**[9,10◦]

**1** Division of Global and Tropical Health, Menzies School of Health Research and Charles Darwin University, Darwin, Northern Territory, Australia, **2** Centre for Epidemiology and Biostatistics, Melbourne School of Population and Global Health, University of Melbourne, Melbourne, Victoria, Australia, **3** Institute for Disease Modeling, Seattle, Washington, United States of America, **4** Victorian Infectious Diseases Reference Laboratory Epidemiology Unit, Royal Melbourne Hospital, University of Melbourne at the Peter Doherty Institute for Infection and Immunity, Melbourne, Victoria, Australia, **5** Foundation for Innovative New Diagnostics (FIND), Geneva, Switzerland, **6** Oxford Big Data Institute, Nuffield Department of Medicine, University of Oxford, Oxford, United Kingdom, **7** Telethon Kids Institute, Perth Children's Hospital, Nedlands, Western Australia, Australia, **8** Curtin University, Bentley, Western Australia, Australia, **9** Nuffield Department of Medicine, Centre for Tropical Medicine and Global Health, University of Oxford, Oxford, United Kingdom, **10** Mahidol Oxford Tropical Medicine Research Unit, Bangkok, Thailand

◦ These authors contributed equally to this work.
* angela.devine@menzies.edu.au

**Data Availability Statement:** The authors confirm that all data underlying the findings and source code needed to reproduce the main results are

## Abstract

### Background

In 2017, an estimated 14 million cases of *Plasmodium vivax* malaria were reported from Asia, Central and South America, and the Horn of Africa. The clinical burden of vivax malaria is largely driven by its ability to form dormant liver stages (hypnozoites) that can reactivate to cause recurrent episodes of malaria. Elimination of both the blood and liver stages of the parasites ("radical cure") is required to achieve a sustained clinical response and prevent ongoing transmission of the parasite. Novel treatment options and point-of-care diagnostics are now available to ensure that radical cure can be administered safely and effectively. We quantified the global economic cost of vivax malaria and estimated the potential cost benefit of a policy of radical cure after testing patients for glucose-6-phosphate dehydrogenase (G6PD) deficiency.

### Methods and findings

Estimates of the healthcare provider and household costs due to vivax malaria were collated and combined with national case estimates for 44 endemic countries in 2017. These provider and household costs were compared with those that would be incurred under 2 scenarios for radical cure following G6PD screening: (1) complete adherence following daily supervised primaquine therapy and (2) unsupervised treatment with an assumed 40% effectiveness. A probabilistic sensitivity analysis generated credible intervals (CrIs) for the estimates. Globally, the annual cost of vivax malaria was US$359 million (95% CrI: US$222 to

available without restriction. All relevant data are within the paper and its Supporting Information files. Code and data can be found at https://github.com/angeladevine/vivax-global-costs .

**Funding:** This work was funded by the Bill & Melinda Gates Foundation (https://www.gatesfoundation.org) to RNP (INV-007122). Funding was also received from the National Health and Medical Research Council of Australia (https://www.nhmrc.gov.au) through the Australian Centre for Research Excellence on Malaria Elimination (ACREME) to JAS, RNP (1134989). RNP is a Wellcome Trust (https://wellcome.org/) Senior Fellow in Clinical Science (200909) and JAS is a National Health and Medical Research Council of Australia (https://www.nhmrc.gov.au) Senior Research Fellow (1104975). PWG is supported by the Bill and Melinda Gates Foundation (https://www.gatesfoundation.org, INV-009390) and the Telethon Trust (https://www.telethon7.com), Western Australia. The funders did not participate in the study design, data collection and analysis, decision to publish, or preparation of the manuscript.

**Competing interests:** The authors have declared that no competing interests exist.

**Abbreviations:** CHEERS, Consolidated Health Economic Evaluation Reporting Standards; CrI, credible interval; G6PD, glucose-6-phosphate dehydrogenase; GDP, gross domestic product; RDT, rapid diagnostic test.

563 million), attributable to 14.2 million cases of vivax malaria in 2017. From a societal perspective, adopting a policy of G6PD deficiency screening and supervision of primaquine to all eligible patients would prevent 6.1 million cases and reduce the global cost of vivax malaria to US$266 million (95% CrI: US$161 to 415 million), although healthcare provider costs would increase by US$39 million. If perfect adherence could be achieved with a single visit, then the global cost would fall further to US$225 million, equivalent to $135 million in cost savings from the baseline global costs. A policy of unsupervised primaquine reduced the cost to US$342 million (95% CrI: US$209 to 532 million) while preventing 2.1 million cases. Limitations of the study include partial availability of country-level cost data and parameter uncertainty for the proportion of patients prescribed primaquine, patient adherence to a full course of primaquine, and effectiveness of primaquine when unsupervised.

## Conclusions

Our modelling study highlights a substantial global economic burden of vivax malaria that could be reduced through investment in safe and effective radical cure achieved by routine screening for G6PD deficiency and supervision of treatment. Novel, low-cost interventions for improving adherence to primaquine to ensure effective radical cure and widespread access to screening for G6PD deficiency will be critical to achieving the timely global elimination of *P. vivax*.

## Author summary

### Why was this study done?

- Vivax malaria is a mosquito-borne febrile illness common in Asia, Latin America, and the Horn of Africa. Although a cause of significant morbidity, its global cost burden has not previously been estimated using country-level data.

- The timely elimination of vivax malaria will require widespread access to safe and effective radical cure.

### What did the researchers do and find?

- Data on the epidemiology and costs of vivax malaria to healthcare providers and households were collated to derive the annual global cost burden of vivax malaria.

- The potential impact of widespread provision of primaquine radical cure was quantified in terms of case incidence and costs for 44 endemic countries.

- The global cost of vivax malaria in 2017 was US$359 million, but this could be reduced by US$93.6 million by screening for G6PD deficiency and provision of supervised high-dose primaquine.

### What do these findings mean?

- Supervised primaquine can lead to a substantial reduction in household and global costs of vivax malaria, although the costs to healthcare providers will be higher.

- Novel, low-cost methods for improving adherence are needed to ensure the affordability and scale-up of radical cure.

## Introduction

Over the last decade, significant gains have been made in reducing the global burden of malaria. Early diagnosis, highly effective antimalarial treatment, and intensive vector control measures have led to a major reduction in the global burden of *Plasmodium falciparum* [1]. The impact of these measures on *Plasmodium vivax*, however, has been more modest. In 2017, vivax malaria was estimated to cause between 13.5 and 15 million cases of malaria [2], with the greatest burden of disease found in remote communities with poor access to healthcare [3]. Outside of sub-Saharan Africa, a rising proportion of malaria is caused by vivax malaria, highlighting the unique challenges in eliminating the parasite [3]. Unlike *P. falciparum*, *P. vivax* forms dormant liver stages (hypnozoites) that reactivate periodically, causing recurrent episodes of malaria (relapses) associated with a cumulative risk of anaemia in addition to direct and indirect attributable mortality [4,5] and ongoing transmission of the parasite [6]. Pregnant women and young children are particularly vulnerable, with vivax malaria causing premature delivery and low birth weight, both of which contribute to perinatal and infant mortality [7–9].

Radical cure of vivax malaria requires a combination of schizontocidal and hypnozoitocidal antimalarial drugs to kill both the blood and liver stages of the parasites. The only widely available antimalarial drug with hypnozoitocidal activity is primaquine, which is usually recommended as a 14-day regimen [10]. Adherence to such a prolonged course of treatment for an acute febrile illness is poor, resulting in a high proportion of patients prescribed unsupervised primaquine in routine clinical practice receiving a dose that is ineffective for radical cure [11,12]. Shorter course treatment regimens offer an alternative strategy that may facilitate greater adherence and more effective antimalarial treatments. Two recent trials have shown that a 7-day regimen of high daily dose primaquine is well tolerated with similar efficacy to the same total dose of primaquine administered over 14 days [13,14]. The licensing of tafenoquine in 2018 provides an alternative hypnozoitocidal drug, which can be administered as a single dose, overcoming the challenge of adherence [15].

Primaquine and tafenoquine are both 8-aminoquinoline compounds and can cause severe haemolysis in individuals with glucose-6-phosphate dehydrogenase (G6PD) deficiency [16]. G6PD deficiency is a common inherited enzymopathy, prevalent in up to 30% of populations residing in malaria-endemic areas [17]. The World Health Organisation recommends that, where possible, individuals should be tested for G6PD deficiency before prescribing primaquine, and this is particularly important when treating patients with shorter high daily dose primaquine regimens, or long-acting tafenoquine.

Concerns regarding severe drug-induced haemolysis and the additional costs of providing G6PD testing frequently result in policy makers and healthcare providers being reluctant to recommend or prescribe radical cure [18]. A large investment has been made in the research

and development of novel point-of-care tests for G6PD deficiency, including qualitative rapid diagnostic tests (RDTs) and quantitative biosensors. These tests are less expensive than the traditional fluorescent spot test [19] and have stimulated interest in their use in areas without laboratory facilities [20], offering new opportunities for improving the management and control of vivax malaria, particularly in remote settings.

Wide-scale adoption of technologies facilitating radical cure of vivax malaria will incur additional costs to providers and funders; whether this represents a worthwhile investment is highly dependent on the global economic impact of vivax malaria, which has yet to be quantified. The aims of this study were to collate information on the costs of illness due to *P. vivax*, quantify the current global economic costs to both healthcare providers and the households of patients, and explore the potential cost–benefit of wide-scale implementation of G6PD screening and primaquine radical cure.

## Methods

### Cases of vivax malaria

The Malaria Atlas Project estimated that the incidence of vivax malaria in 2017 was 14.2 million cases across the 44 endemic countries included in this analysis (S1 Table) [2]. These estimates refer to symptomatic vivax malaria and were used as the time horizon for the costs from the healthcare provider and societal perspectives. Estimates utilise treatment-seeking rates at public facilities to adjust for cases that would not be included in national reporting systems due to individuals attending private healthcare providers or never seeking treatment. National estimates of treatment-seeking behaviour were derived from household survey data [21] that were categorised according to whether patients sought treatment with any provider (including public or private healthcare providers, pharmacies, or shops) or did not seek treatment outside of their own home. Treatment-seeking values were modelled for countries and years without household data using socioeconomic indicator variables and a Gaussian process regression [2]. Case values for 2017 were also adjusted for reporting completeness using subnational values publicly available from country programmes or national values as reported in the World Malaria Report [22]. Age-specific incidence rates were derived from a model originally calibrated for *P. falciparum* but adapted for *P. vivax* [23,24]. Case estimates for 2017 were available for all endemic countries, except for the majority of sub-Saharan Africa due to a paucity of case data. Those for North Korea were excluded from the analysis due to a scarcity of complementary cost data. This study is reported as per the Consolidated Health Economic Evaluation Reporting Standards (CHEERS) guidelines (S1 CHEERS Checklist).

### Costs to healthcare providers

For patients seeking treatment at healthcare providers, the proportion of malaria cases diagnosed by either RDT or microscopy as well as the drugs prescribed in 2017 were derived for each country from the World Malaria Report (S1 Table) [22]. After applying the percent of cases confirmed by diagnostic test to the population seeking treatment, the percent of confirmed cases diagnosed by RDT was used to calculate the RDT costs, while the cost of microscopy was applied to the remaining individuals. Madagascar did not report vivax-specific antimalarial treatments, so it was assumed that the species of infection was not distinguished and that patients with *P. vivax* were treated with the same antimalarials as patients with uncomplicated confirmed *P. falciparum*. In the 42 countries in which primaquine was recommended in national guidelines [22], this was assumed to be in line with the WHO Antimalarial Treatment Guidelines, in which treatment is only recommended to nonpregnant and nonlactating females and children over the age of 1 year [10]. Accordingly, the estimated proportion

of patients who were pregnant or lactating were excluded from those over the age of 15 [25], and 20% of cases under the age of 5 were excluded from primaquine eligibility. From experience in the field, provider compliance to national treatment guidelines for eligible patients was assumed to be 40%, and this was applied to the eligible population to determine the number and associated cost of primaquine prescriptions. In 2017, only Malaysia routinely assessed G6PD status prior to primaquine administration; accordingly, the cost of a fluorescent spot test was applied to all patients eligible for primaquine, and the cost of primaquine was added for the proportion who were G6PD normal. The prevalence of the population with G6PD deficiency (<30% activity) [17] and proportion of females who were pregnant or lactating [25] are listed in S1 Table. Diagnostic tests, treatment, days lost to illness, and case estimates along with all country-level assumptions are shown in S1 Table.

Where possible, costs were collected in local currencies and inflated to 2017 using gross domestic product (GDP) deflators [26] before converting to United States Dollars (US$) using 2017 exchange rates [27]. The US GDP deflator was used for missing years for Djibouti, Eritrea, and Venezuela. Overhead treatment costs were taken from WHO-CHOICE [28] and supplemented with drug costs from the International Medical Products Price Guide [29]. Since the majority of vivax malaria occurs in rural areas, the cost of a primary care health centre without beds was used as the cost of outpatient visits, and the cost per bed day in a primary-level hospital was used for inpatient visits [28]. For Somalia, where healthcare costs were unavailable, these were derived from neighbouring Ethiopia. It was assumed that 2% of patients who sought treatment for malaria required hospitalization [30] and that these required 3 days of inpatient care [31,32]. Diagnostic test costs for vivax malaria were obtained from the literature [19,33,34] and applied by WHO region (S2 Table).

## Costs to households

Direct costs to the patients included treatment, transportation, and any previous treatment seeking for those who sought treatment at multiple locations [19]. Direct costs were only applied to those seeking treatment, whereas indirect costs were applied to all cases. Indirect costs included the cost of the days during which patients were unable to attend to their usual activities due to illness and days when a caregiver was required to stop doing usual activities to care for a patient with vivax malaria. The number of days lost due to illness for patients and carers was taken from the weighted average number of patients in each WHO region (S1 Table) [19]. For children under the age of 5, only the carer days lost were applied to cases. The days lost due to illness were valued at 1 GDP per capita per day [35,36]. An overview of all parameters, assumptions, and data sources can be found in S2 File.

## Cost–benefit of global implementation of G6PD testing with radical cure

In addition to the baseline global costs, 2 scenario analyses were explored to quantify the potential impact of the global implementation of a policy in which high-dose primaquine radical cure (total dose of 7 mg/kg) was administered to eligible patients after screening for G6PD deficiency. In both scenarios, the cost of a high total dose of primaquine (7 mg/kg) was used, given its high efficacy across multiple and diverse locations [13,14]; however, this potentially overestimates the cost of primaquine in areas where a lower total dose may have sufficed [37]. In the first scenario, *Supervised radical cure*, it was assumed that daily supervision of primaquine therapy administered to eligible patients would result in perfect adherence. The cost of treatment supervision was estimated to be 1.6 healthcare worker days (1 hour per day for 13 days assuming a work day of 8 hours). Since most community healthcare workers are unpaid, their time was valued using the GDP per capita per day [35,36]. In the second scenario,

*Unsupervised radical cure*, primaquine radical cure in G6PD normal patients was assumed to have only 40% effectiveness due to lower patient adherence. Limited information is available on adherence to and effectiveness of a 14-day primaquine regimen [11,38]. In both scenarios, G6PD screening was assumed to be undertaken using a qualitative lateral flow RDT that identifies individuals with enzyme activity below 30% but does not identify heterozygous females with intermediate enzyme activity [20]. The G6PD RDT was assumed to have a 96% sensitivity, which was derived from a recent meta-analysis [20].

A cost–benefit analysis was then carried out in which these 2 scenarios were compared with the baseline global costs to assess the changes in global incidence and costs. The number of cases averted was determined from the number of treatment-seeking individuals in the baseline global cost estimates who were G6PD normal and eligible to receive primaquine, multiplied by the proportion of relapses that would be averted. In the *Supervised radical cure* scenario, high-dose primaquine reduced the risk of relapse by 88% in patients completing treatment [39]. For the *Unsupervised radical cure* scenario, the effectiveness was assumed to remain at 40%. In countries where primaquine is already prescribed, the percent of relapses prevented by treatment with effective primaquine radical cure in the baseline global costs (i.e., the proportion prescribed primaquine multiplied by the proportion receiving an effective dose) were subtracted from the proportion of relapses averted in each radical cure scenario. The cases averted were then subtracted from the global cost incidence before calculating treatment-seeking behaviour and associated costs.

The assignment of costs to patients with vivax malaria is presented in Fig 1. Unit costs were taken from the literature review and applied by WHO region [19,33,34]. The cost of a G6PD RDT, including an additional blood draw, was applied to the patient population eligible to

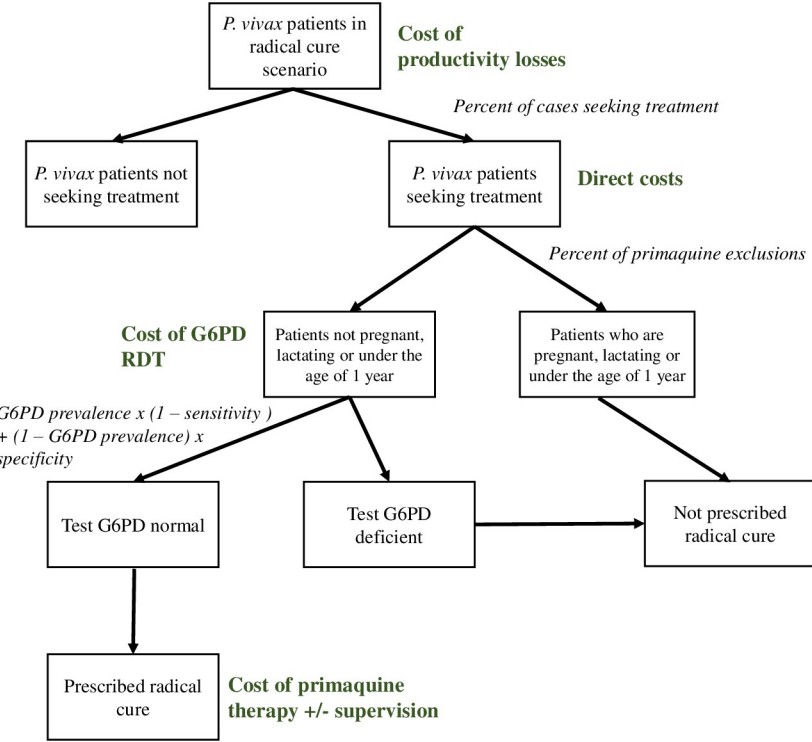

**Fig 1. Flow diagram for the application of costs in the radical cure scenarios.** Of those prescribed radical cure, only those who are G6PD normal are able to have an effective dose.

receive primaquine (i.e., not pregnant, lactating, or under the age of 1 year). The cost of the G6PD RDT in SEARO was the average of the other regions since the cost from Indonesia was exceptionally high and was only applied within the country. In Malaysia, it was assumed that the fluorescent spot test would continue to be used for G6PD diagnosis. These costs replaced any preexisting costs of screening and primaquine administration used in the global cost analysis.

### Sensitivity analyses

The impact of excluding productivity losses for patients under the age of 15 was explored in a sensitivity analysis for the baseline global costs. For the *Supervised radical cure* scenario, a sensitivity analysis was used to explore the impact of reducing the number of days of supervision from 13 days for a fully supervised 14-day primaquine regimen to 6 visits for a fully supervised 7-day regimen, and to 1 visit for a review at day 7 of a 7-day primaquine regimen. In these scenarios, it was assumed that adherence would remain at 100%. A second one-way sensitivity analysis varied the percent of recurrent cases prevented by a full course of high-dose primaquine from 82% to 92% [39]. For the *Unsupervised radical cure* scenario, a one-way sensitivity analysis used a range of 30% [11] to 60% [40] to quantify the impact of the effectiveness of unsupervised primaquine.

Credible intervals (CrIs) were calculated for the global cost and scenario estimates through a probabilistic sensitivity analysis. The probabilistic sensitivity analysis drew 10,000 samples from ranges around the base case values. Where possible, these were the reported 95% confidence intervals, but otherwise plausible ranges were used (S1 and S2 Tables and S2 File). Cost parameters were given gamma distributions, proportions were given beta distributions, and the incidence a normal distribution (S2 File). The limits of the 95% CrIs are the 2.5th and 97.5th percentiles of the 10,000 samples.

### Online model

In view of the uncertainty around the model parameters and their marked heterogeneity between endemic areas, a web-based model was developed with options to vary key parameters for the baseline global costs and a radical cure scenario for each country. The model provides the option of including costs due to primaquine-induced haemolysis, which were not included in the primary analysis due to uncertainty on the frequency and associated direct costs.

## Results

### Global costs due to vivax malaria

The age model stratified the 14.2 million *P. vivax* cases in 2017 into 7.1 million (49.8%) in adults aged 15 and older, 3.5 million (25.0%) in children aged 5 to 14, and 3.6 million (25.2%) in infants less than 5 years old (Table 1). Of the 5.3 million treatment-seeking adults, 166,144 (3.2%) were estimated to be pregnant or lactating. Overall, 884,000 patients over the age of 1 year who sought treatment had severe (<30%) G6PD deficiency (Table 1). Of the 10.5 million patients with vivax malaria who sought treatment, 3.8 million (37%) were prescribed primaquine, and 1.5 million (15%) received an effective antirelapse dose (S4 Table).

The estimated baseline global cost of vivax malaria in 2017 was US$359 million (95% CrI: US$222 to 563 million; Table 2). The cost burden varied widely between countries, which largely reflected the underlying case estimates (Fig 2A). India carried the greatest cost burden of US$175 million (95% CrI: US$97 to 298 million), accounting for 49% of the total global cost (Table 2). Other high-cost countries were Pakistan (US$60 million), Venezuela (US$42

**Table 1. Demographics and case numbers of patients with vivax malaria.**

| | WHO Region | | | | | Total |
|---|---|---|---|---|---|---|
| | **AFRO** | **EMRO** | **PAHO** | **SEARO** | **WPRO** | |
| **Population at risk** | 127,000,500 | 375,944,496 | 315,466,914 | 1,649,572,032 | 1,524,044,616 | 3,992,028,558 |
| **Number of patients with vivax malaria** | | | | | | |
| Infants (0–4 years old) | 217,490 | 1,487,805 | 183,627 | 1,498,725 | 184,930 | 3,572,577 |
| Children (5–14 years old) | 187,122 | 1,388,478 | 170,783 | 1,647,335 | 146,117 | 3,539,835 |
| Adults (15+) | 271,038 | 2,128,215 | 424,959 | 4,011,290 | 217,856 | 7,053,358 |
| **Population seeking treatment** | | | | | | |
| Infants (0–4 years old) | 66,607 | 1,134,049 | 118,102 | 1,165,419 | 129,553 | 2,613,730 |
| Children (5–14 years old) | 56,892 | 1,063,808 | 108,927 | 1,280,834 | 102,379 | 2,612,840 |
| Adults (15+) | 82,838 | 1,643,540 | 270,185 | 3,117,024 | 153,054 | 5,266,641 |
| Pregnant or lactating females | 2,930 | 44,430 | 4,838 | 109,141 | 4,805 | 166,144 |
| Patients with severe G6PD deficiency (<30% activity) in those >1 year old | 9,380 | 463,867 | 25,598 | 354,864 | 30,044 | 883,753 |
| Number eligible for primaquine | 172,175 | 2,974,714 | 422,248 | 4,643,713 | 309,587 | 8,522,437 |

AFRO, Africa Region; EMRO, Eastern Mediterranean Region; G6PD, glucose-6-phosphate dehydrogenase; PAHO, Americas Region; SEARO, Southeast Asia Region; WHO, World Health Organisation; WPRO, Western Pacific Region.

million), Indonesia (US$21 million), Brazil (US$18 million), Papua New Guinea (US$9 million), and Sudan (US$7 million). While Ethiopia has the third highest case burden, it was ninth in cost burden, which was driven by the low numbers seeking treatment and a low GDP per capita per day.

Overall, 70% (US$252 million) of the global cost burden was attributable to indirect household costs, 18% (US$64 million) to healthcare provider costs, and 12% (US$43 million) to direct household costs. In the sensitivity analysis where only productivity losses to adults were included, the global cost decreased to US$303 million (Table 3 and S3 Table).

### *Supervised radical cure* scenario

In this scenario, the total number of cases would decrease from 14.2 million to 8.0 million, a 43% reduction (6.1 million cases) in 2017 (Table 4). Of the 5.8 million people seeking treatment, 4.7 million (81%) were prescribed radical cure (including G6PD deficient with false negative test results), of whom 2.4 million were adults, 1.2 million children, and 1.1 million infants (S4 Table). Approximately 19,000 patients with severe G6PD deficiency would have been treated with high-dose primaquine due to the RDT providing false normal results.

The additional provider costs of delivering this scenario were US$39.4 million, increasing the total from US$63.6 million to US$103 million. The total provider costs consisted of US$20.5 million (20%) for G6PD screening, US$44.5 million (43%) for primaquine supervision, and US$38.0 (37%) for case management (S3 Table). While the total provider costs increased, household costs decreased by US$133 million (from US$296 million to US$163 million; Fig 3 and Table 3). Overall, the global cost of vivax malaria in this scenario was US$266 million, representing $94 million in cost savings from the baseline global costs (Fig 2B and Table 3). When varying the bounds of vivax malaria recurrences preventable with a full course of high dose from 88% to 82% and 92%, the global cost of the *Supervised radical cure* scenario ranged from $284 million to US$254 million, respectively (Table 3).

Reducing the number of supervision visits to 6 (equivalent to a fully supervised 7-day primaquine regimen), decreased provider costs by US$23.9 million (23%), from US$103 million to US$79.1 million (Fig 3 and Table 3). Further reducing the supervision visits to 1 visit,

**Table 2. Baseline global costs for vivax malaria by country.** Costs are in 2017 United States Dollars. The provider cost per case can be found in S3 Table.

| Country | Provider Costs | Direct Household Costs | Indirect Household Costs | Total Cost | 95% CrI |
|---|---|---|---|---|---|
| Afghanistan | 824,207 | 1,096,830 | 1,695,390 | 3,616,427 | 2,521,907–5,004,143 |
| Bangladesh | 3,857 | 2,823 | 20,124 | 26,804 | 14,012–47,721 |
| Belize | 85 | 71 | 266 | 425 | 294–539 |
| Bhutan | 136 | 68 | 775 | 979 | 495–1,803 |
| Bolivia | 73,923 | 90,951 | 306,228 | 471,101 | 329,167–639,141 |
| Brazil | 724,969 | 1,068,019 | 15,790,075 | 17,583,063 | 12,045,680–24,355,381 |
| Cambodia | 108,770 | 53,939 | 198,902 | 361,611 | 255,693–506,171 |
| China | 173 | 43 | 1,231 | 1,447 | 869–2,203 |
| Colombia | 530,699 | 360,762 | 2,450,830 | 3,342,292 | 2,463,464–4,381,196 |
| Djibouti | 2,292 | 1,499 | 13,231 | 17,020 | 2,076–84,155 |
| Ecuador | 22,674 | 19,008 | 95,512 | 137,194 | 98,542–181,746 |
| El Salvador | 197 | 188 | 598 | 983 | 351–1,682 |
| Eritrea | 32,177 | 15,596 | 151,114 | 198,888 | 116,137–312,887 |
| Ethiopia | 655,548 | 426,961 | 3,863,726 | 4,946,235 | 2,145,863–8,682,535 |
| Guatemala | 124,298 | 120,874 | 492,056 | 737,227 | 508,741–1,013,772 |
| Guyana | 134,264 | 177,508 | 719,168 | 1,030,939 | 725,173–1,385,411 |
| Honduras | 37,382 | 46,065 | 104,626 | 188,073 | 145,054–238,013 |
| India | 31,312,534 | 19,423,275 | 124,012,650 | 174,748,458 | 97,160,933–298,000,840 |
| Indonesia | 4,663,828 | 1,433,060 | 15,392,969 | 21,489,856 | 12,078,025–37,452,273 |
| Iran | 155 | 55 | 877 | 1,089 | 95–3,388 |
| Laos | 101,272 | 46,828 | 371,566 | 519,666 | 278,947–852,076 |
| Madagascar | 288,226 | 135,187 | 412,356 | 835,769 | 540,315–1,210,461 |
| Malaysia | 53,410 | 4,208 | 148,344 | 205,962 | 127,749–315,045 |
| Mexico | 16,589 | 10,023 | 69,365 | 95,976 | 10,435–253,163 |
| Myanmar | 273,327 | 259,050 | 1,269,994 | 1,802,371 | 972,863–3,179,252 |
| Nepal | 9,479 | 11,586 | 36,990 | 58,056 | 32,660–98,397 |
| Nicaragua | 53,897 | 81,763 | 156,557 | 292,218 | 217,508–379,777 |
| Pakistan | 9,848,148 | 11,234,363 | 38,929,199 | 60,011,710 | 43,879,014–81,229,210 |
| Panama | 21,607 | 12,024 | 140,886 | 174,516 | 126,566–232,988 |
| Papua New Guinea | 1,526,310 | 821,617 | 6,814,993 | 9,162,922 | 5,367,104–14,668,561 |
| Peru | 871,144 | 722,995 | 4,052,541 | 5,646,681 | 4,290,355–7,216,415 |
| Philippines | 34,222 | 15,048 | 202,708 | 251,978 | 144,281–411,933 |
| Saudi Arabia | 3,332 | 311 | 18,182 | 21,826 | 6,347–41,916 |
| Solomon Islands | 297,597 | 118,857 | 768,318 | 1,184,772 | 682,891–1,902,202 |
| Somalia | 11,328 | 15,575 | 5,253 | 32,156 | 14,785–52,234 |
| South Korea | 20,175 | 1,294 | 147,273 | 168,742 | 101,814–259,931 |
| Sudan | 2,200,438 | 1,080,125 | 3,619,371 | 6,899,934 | 3,512,411–10,953,919 |
| Suriname | 1,312 | 1,267 | 6,202 | 8,782 | 6,454–11,525 |
| Thailand | 34,757 | 10,548 | 264,009 | 309,315 | 152,154–569,041 |
| Timor-Leste | 66 | 42 | 185 | 294 | 170–490 |
| Vanuatu | 19,478 | 5,748 | 56,016 | 81,243 | 49,636–125,818 |
| Venezuela | 8,622,285 | 4,448,362 | 29,359,804 | 42,430,450 | 30,629,909–56,094,833 |
| Vietnam | 18,008 | 10,379 | 103,435 | 131,822 | 77,289–209,172 |
| Yemen | 33,172 | 16,130 | 42,431 | 91,733 | 67,572–121,878 |
| Total | 63,611,747 | 43,400,925 | 252,306,326 | 359,319,005 | 221,901,800–562,685,237 |

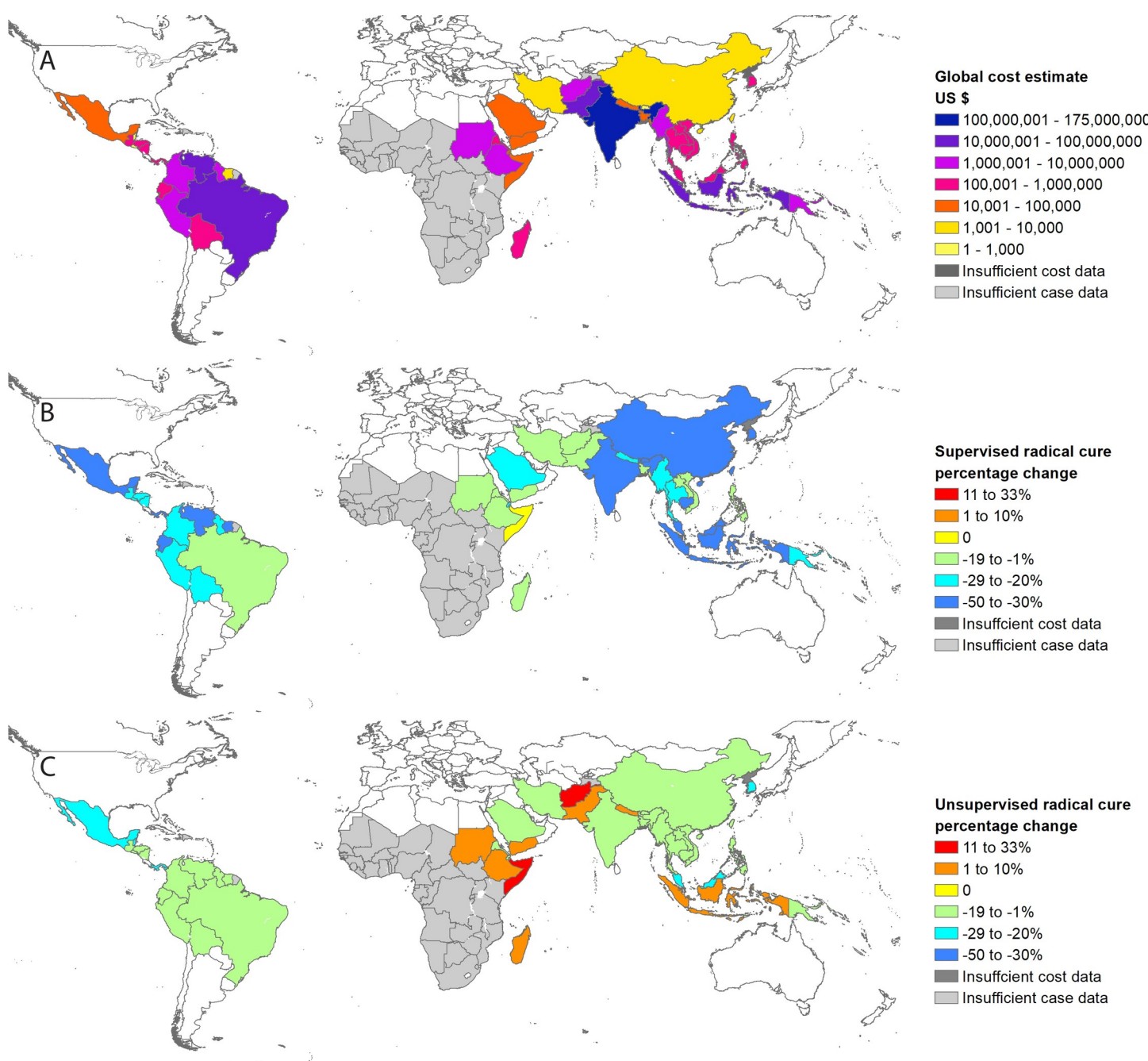

**Fig 2. Global map of the economic cost burden due to vivax malaria and potential impact of radical cure.** (A) The baseline global costs, (B) the *Supervised radical cure* scenario, and (C) the *Unsupervised radical cure* scenario. Percentage change in total costs from the baseline global costs are shown for the radical cure scenarios. Costs are in 2017 United States Dollars. Countries in light grey are thought to have endemic *P. vivax* but insufficient information to generate case estimates. Countries in dark grey have insufficient cost data. Global national shapefile obtained from the Malaria Atlas Project (MAP; https://malariaatlas.org/) and available for download through the malariaAtlas R package.

decreased provider costs by a further US$17.1 million to US$61.9 million, US$1.6 million less than the provider costs in the baseline global costs (Fig 3 and Table 3). If adherence could be achieved with a single visit, it would result in $135 million in cost savings from the baseline global costs.

**Table 3. Results of the baseline global costs and *Supervised radical cure* and *Unsupervised radical cure* scenarios with 95% credible intervals for the baseline total cost estimates.** One-way sensitivity analyses to 6 visits and 1 visit of supervision as compared to 13 visits. All costs are in 2017 United States Dollars and rounded to the nearest 1,000.

| Cost component | Baseline global costs | | *Supervised radical cure* scenario | | | | | | *Unsupervised radical cure* scenario | | |
|---|---|---|---|---|---|---|---|---|---|---|---|
| | Results | One-way SA | Results | One-way SA | | | | | Results | One-way SA | |
| | | Excluding productivity losses in children | | Six supervision visits | One supervision visit | Low proportion cases prevented by full PQ course[a] | High proportion cases prevented by full PQ course[a] | | | Low effectiveness of PQ without supervision[b] | High effectiveness of PQ without supervision[b] |
| Total healthcare provider costs | 63,612,000 | 63,612,000 | 103,043,000 | 79,059,000 | 61,927,000 | 110,042,000 | 98,377,000 | | 90,389,000 | 102,404,000 | 78,375,000 |
| Total household costs | 295,707,000 | 239,273,000 | 162,676,000 | 162,676,000 | 162,676,000 | 173,745,000 | 155,297,000 | | 251,232,000 | 284,588,000 | 217,876,000 |
| Direct | 43,401,000 | 43,401,000 | 24,286,000 | 24,286,000 | 24,286,000 | 25,877,000 | 23,225,000 | | 37,013,000 | 41,804,000 | 32,222,000 |
| Indirect | 252,306,000 | 195,872,000 | 138,390,000 | 138,390,000 | 138,390,000 | 147,869,000 | 132,072,000 | | 214,219,000 | 242,784,000 | 185,654,000 |
| Total costs (95% CrIs) | 359,319,000 (221,902,000–562,685,000) | 302,885,000 | 265,719,000 (160,996,000–415,443,000) | 241,735,000 | 224,603,000 | 283,788,000 | 253,674,000 | | 341,621,000 (208,558,000–532,457,000) | 386,993,000 | 296,251,000 |

CrIs, credible intervals; PQ, primaquine; SA, sensitivity analysis.

[a]Varied from 0.88 to 0.82 for low value and 0.92 for high value.

[b]Varied from 0.40 to 0.10 for low value and 0.70 for high value.

### *Unsupervised radical cure* scenario

In this scenario, the impact on the total global incidence and cost was modest, with the number of cases decreasing by 2.1 million to 12.1 million (Table 4); this is 4.1 million more cases than in the *Supervised radical cure* scenario. When the wide bounds used for the effectiveness of primaquine without supervision were varied from 40% to 10% and 70% effectiveness, it resulted in a change of 1.5 million cases in both directions. The corresponding variation in costs was US$387 million when assuming 10% effectiveness and US$296 million with 70% effectiveness.

The additional intervention costs under the *Unsupervised radical cure* scenario were entirely attributable to the provision of G6PD testing and resulted in an increase in provider costs of US$26.8 million to US$90.4 million. Conversely, household costs decreased by US$44.5 million (15%) to US$251 million (Table 3). The provider costs were higher than the baseline global costs for all countries. The total cost of vivax malaria from a societal perspective decreased by US$17.7 million to US$342 million (Figs 2C and 3, Table 3).

## Discussion

To our knowledge, this paper collates for the first time the available country-level data on the epidemiology and costs of vivax malaria and estimates the associated global economic burden. The total global cost of vivax malaria in 2017 was estimated to be US$359 million, of which 82% was incurred by households and 18% by healthcare providers. The first scenario exploring how global costs would change with universal access to supervised radical cure following G6PD testing highlights that healthcare provider costs could nearly double while household costs could fall by almost a half, leading to cost savings of US$93.6 million and the prevention of 6.1 million malaria cases. The alternative scenario of G6PD testing prior to prescribing

**Table 4. Annual incidence of vivax malaria and numbers seeking treatment for the baseline global cost estimates, and annual incidence and percent reduction from the baseline estimates for the *Supervised radical cure* and *Unsupervised radical cure* scenarios.**

| Country | Baseline incidence | Baseline treatment seeking | *Supervised radical cure* incidence | Percent reduction from baseline[a] | *Unsupervised radical cure* incidence | Percent reduction from baseline[a] |
|---|---|---|---|---|---|---|
| Afghanistan | 492,579 | 313,380 | 308,337 | 37% | 431,165 | 12% |
| Bangladesh | 1,333 | 743 | 865 | 35% | 1,178 | 12% |
| Belize | 6 | 5 | 2 | 67% | 5 | 17% |
| Bhutan | 24 | 18 | 13 | 46% | 21 | 13% |
| Bolivia | 10,926 | 6,316 | 6,887 | 37% | 9,580 | 12% |
| Brazil | 186,014 | 74,168 | 139,934 | 25% | 170,654 | 8% |
| Cambodia | 21,814 | 19,264 | 11,047 | 49% | 18,225 | 16% |
| China | 18 | 15 | 9 | 50% | 16 | 11% |
| Colombia | 42,622 | 25,053 | 26,900 | 37% | 37,381 | 12% |
| Djibouti | 655 | 428 | 381 | 42% | 564 | 14% |
| Ecuador | 1,733 | 1,320 | 905 | 48% | 1,457 | 16% |
| El Salvador | 17 | 13 | 9 | 47% | 15 | 12% |
| Eritrea | 9,921 | 5,570 | 6,515 | 34% | 8,785 | 11% |
| Ethiopia | 573,729 | 152,486 | 457,125 | 20% | 520,727 | 9% |
| Guatemala | 13,725 | 8,394 | 8,490 | 38% | 11,980 | 13% |
| Guyana | 18,661 | 12,327 | 10,901 | 42% | 16,074 | 14% |
| Honduras | 5,081 | 3,199 | 3,071 | 40% | 4,411 | 13% |
| India | 6,612,425 | 5,111,388 | 3,553,457 | 46% | 5,592,769 | 15% |
| Indonesia | 429,941 | 377,121 | 201,358 | 53% | 353,746 | 18% |
| Iran | 22 | 16 | 14 | 36% | 20 | 9% |
| Laos | 23,870 | 16,724 | 14,715 | 38% | 20,818 | 13% |
| Madagascar | 92,000 | 48,281 | 61,576 | 33% | 78,171 | 15% |
| Malaysia | 2,017 | 1,503 | 1,102 | 45% | 1,711 | 15% |
| Mexico | 863 | 696 | 416 | 52% | 713 | 17% |
| Myanmar | 105,458 | 68,171 | 63,551 | 40% | 91,489 | 13% |
| Nepal | 4,239 | 3,049 | 2,344 | 45% | 3,607 | 15% |
| Nicaragua | 8,458 | 5,678 | 4,830 | 43% | 7,249 | 14% |
| Pakistan | 3,993,746 | 3,209,818 | 2,223,848 | 44% | 3,403,781 | 15% |
| Panama | 1,083 | 835 | 547 | 49% | 904 | 17% |
| Papua New Guinea | 418,872 | 293,435 | 245,794 | 41% | 361,180 | 14% |
| Peru | 75,085 | 50,208 | 43,101 | 43% | 64,423 | 14% |
| Philippines | 10,002 | 5,374 | 6,638 | 34% | 8,880 | 11% |
| Saudi Arabia | 117 | 89 | 65 | 44% | 100 | 15% |
| Solomon Islands | 62,766 | 42,449 | 41,760 | 33% | 55,763 | 11% |
| Somalia | 7,646 | 4,450 | 4,903 | 36% | 6,731 | 12% |
| South Korea | 608 | 462 | 301 | 50% | 506 | 17% |
| Sudan | 502,472 | 308,607 | 333,917 | 34% | 446,287 | 11% |
| Suriname | 122 | 88 | 65 | 47% | 102 | 16% |
| Thailand | 3,915 | 2,776 | 2,292 | 41% | 3,374 | 14% |
| Timor-Leste | 15 | 11 | 8 | 47% | 13 | 13% |
| Vanuatu | 2,903 | 2,053 | 1,691 | 42% | 2,498 | 14% |
| Venezuela | 414,973 | 308,914 | 230,023 | 45% | 353,323 | 15% |
| Vietnam | 6,033 | 3,707 | 3,794 | 37% | 5,287 | 12% |
| Yemen | 7,261 | 4,609 | 4,437 | 39% | 6,320 | 13% |
| Total | 14,165,770 | 10,493,211 | 8,027,938 | 43% | 12,102,003 | 15% |

[a]Equations describing the calculation of these can be found in S1 File.

unsupervised primaquine could increase healthcare provider costs by 42%, but decrease household costs by only 15%, while preventing 2.1 million cases of vivax malaria. The overall cost savings in this scenario were reduced to US$17.7 million. Although realistically these changes would take time and resources to scale up and reap the benefits, the 2 scenarios provide useful insights into the potential impact of a policy and widespread implementation of G6PD screening and radical cure.

Our analysis estimates the current global societal costs of vivax malaria and the prospects for reducing these costs if radical cure strategies were to be widely implemented; it is not, however, a comprehensive cost-effectiveness analysis for the introduction of radical cure strategies and programmes, which would require consideration of other factors and further contextual and country-specific adaptations. Changing antimalarial policy and practice, for instance, would incur further investment in implementation activities, including training and strengthening supply lines, and these costs can be substantial [41,42], and should be included in the country-level cost-effectiveness analyses. The increased healthcare provider costs that we describe alongside the additional resources needed to implement the policy changes might present a major challenge for sustained financing and a disincentive to changing national policy. To put this in context, however, the additional US$39 million provider expenses required for global G6PD screening and treatment represents only 1% of the US$3.1 billion spent on

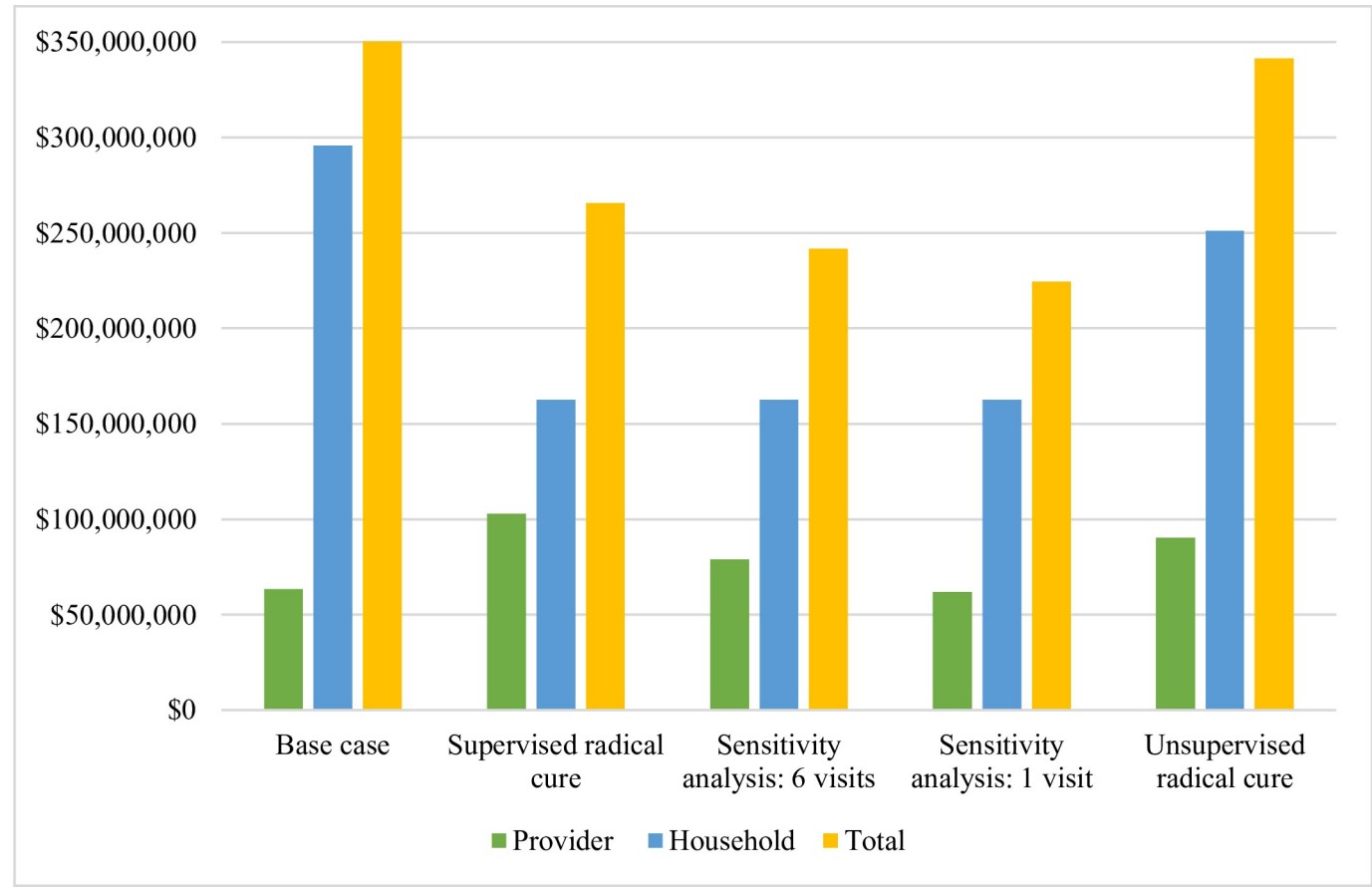

**Fig 3. Comparison of provider, household, and total cost comparison of the baseline global costs and the *Supervised radical cure* and *Unsupervised radical cure* scenarios.** Sensitivity analyses for the *Supervised radical cure* scenario included 6 visits and 1 visit of supervision as compared to 13 visits. Costs are in 2017 United States Dollars.

malaria control activities in 2017, an investment that could potentially halve the global burden of *P. vivax*. Since a large proportion of vivax malaria episodes are attributable to relapses [43,44], investment in safe and effective radical cure will be crucial to achieving the timely elimination of vivax malaria.

Adherence to a complete course of primaquine remains a key obstacle in implementing effective radical cure even for the 7-day course [45], and yet little research has been undertaken to design suitable options to address this. Potential solutions include supervised administration of tablets (as presented in our analysis), or potentially lower cost solutions such as phone calls or text messages, and educational programmes for healthcare workers, patients, and communities [12,46]. The sensitivity analysis highlighted that if high adherence could be achieved with a single visit, then the provider costs would be similar to the baseline global costs. Timely patient review provides an important opportunity to identify drug-related adverse effects, such as gastrointestinal upset or primaquine-induced haemolysis early, so that medication can be stopped and further deterioration prevented. The introduction of single-dose tafenoquine provides another alternative that avoids the challenges of compromised adherence; however, this will require more stringent and costly diagnosis of G6PD deficiency with a quantitative test to exclude treatment of individuals with intermediate or severe deficiency (<70% enzyme activity). Routine quantitative G6PD testing requires hand-held devices to be placed at healthcare facilities, adding significant provider costs. Since these costs will vary considerably with patient throughputs and which levels of healthcare facilities the devices are utilised at, we were unable to include them in our analysis. Until tafenoquine and quantitative testing become widely available, primaquine will continue to be the standard of care; and thus complementary interventions to improve adherence will be critical to malaria elimination efforts.

The scenario analyses focused on the reduction of recurrent infections due to relapsing infections and do not take into consideration the impact on transmission, which can be substantial [47]. Recent estimates suggest that over 70% of recurrent infections are likely to be due to relapsing infections [39]; these constitute a major determinant of transmission, sustaining endemicity over seasonal fluctuations in vector numbers [48]. Furthermore, since recurrent episodes of vivax malaria can result in a cumulative risk of severe anaemia and its associated morbidity and mortality, implementation of effective radical cure is likely to have both direct benefits (i.e., case reductions) as well as indirect benefits by reducing hospitalization and clinic encounters associated with increased susceptibility to other comorbidities [4]. These factors imply that we have likely underestimated the benefits of radical cure.

While we did not attempt to capture the cost of deaths due to vivax malaria, we did include the cost of time lost to illness [19]. The inclusion and valuation of productivity losses, or costs associated with inability to work or participate in leisure activities due to illness or death, is challenging, particularly in individuals who would not be receiving a wage for their usual activities. Estimates of GDP per capita per day were applied to carers for all cases, but only to patients older than 5 years of age, in order to valuate productivity losses for adults and educational impact for children. Restricting patient days lost in the baseline global costs to adults 15 years and older reduced productivity losses by US$56 million. It should be noted, however, that these calculations do not attempt to account for wider long-term economic impacts of disease, such as school performance [49,50], decreased fertility [51], and labor productivity [52].

Our study has a number of important limitations. A key determinant of the global cost was the national estimates of vivax malaria cases, which varied significantly due to the quality of national reporting and treatment-seeking practices. The case estimates from 3 countries with the highest economic burden of vivax malaria (India, Pakistan, and Venezuela) have been inflated from the nationally reported data to reflect reporting completeness; these adjustments are necessary but introduce further uncertainty into the analysis. Case counts are scaled up

based on the estimated treatment-seeking rates in each country. The rate of seeking care and percentage of this which occurs through facilities that are integrated into the health management information systems varies widely between vivax-endemic countries [53]. The age-specific case estimates were obtained from a model developed for falciparum malaria [23,24]. As more age-specific data become available through digital platforms for managing routine surveillance data, this model could be recalibrated to better reflect the epidemiology of vivax malaria in the future.

While most parameters will vary across different endemic settings, estimates are often imprecise and only available from a few locations. In the *Unsupervised radical cure* scenario, effectiveness was a key determinant with a range of 10% [11] to 70%. Another critical factor that was not accounted for in our analysis was the proportion of healthcare providers who prescribe primaquine to vivax malaria patients where the treatment regimen is recommended in national antimalarial guidelines. This will be influenced by a range of factors including supply chain, cost, and fear of causing primaquine-induced haemolysis in areas where G6PD testing is unavailable [18]. The scenario analyses only included costs over a 1-year time horizon; accordingly, relapses prevented beyond the time frame are not captured, thus underestimating the cost savings. The cost of scale-up required to achieve provider compliance with G6PD screening and radical cure are also not included, underestimating the cost of implementation. Furthermore, the long-term effects are likely to fluctuate over time, particularly as countries near elimination and cases become rare events.

Costs specific to vivax malaria vary widely between countries but, in view of the sparse data, cost estimates had to be extrapolated regionally. Public provider costs were applied to all individuals seeking treatment, reflecting the economic cost of treatment, while patient costs would likely be higher when seeking treatment at private providers. Furthermore, relapse patterns can vary considerably within and between countries, particularly high burden and geographically diverse countries such as India and Indonesia, impacting the costs and benefits of radical cure. Finally, the costs of primaquine-induced haemolysis were not factored into the analysis, since these were assumed to be relatively rare and have significant variability in their frequency and severity [16]. To address these uncertainties and facilitate investigation of individual country scenarios, an online application is provided, so that these parameters can be varied and their impact on costs explored (http://lab.qmalaria.org/shiny/appPVcost/). As further data on these parameters are collected and their bounds determined, the certainty of the global cost burden estimates will improve significantly.

In conclusion, our analysis highlights the substantial global economic burden of vivax malaria, which is driven primarily by direct household costs and productivity losses. Provision of safe and effective radical cure is possible but will require an increased investment that could be a disincentive to national malaria control programmes. Our findings suggest that such an investment could ensure high antirelapse effectiveness with substantial cost savings at the societal level and reductions in malaria case numbers. Novel point-of-care G6PD tests are now available along with short-course radical cure regimens such as 7-day primaquine regimen and tafenoquine, which will improve adherence and effectiveness substantially [14,15,54]. Widespread safe and effective radical cure after screening for G6PD deficiency presents a critical challenge for the management of vivax malaria; quantifying the costs and outcomes associated with this treatment will pave the way to the ultimate elimination of the parasite.

## Supporting information

**S1 CHEERS Checklist. Consolidated Health Economic Evaluation Reporting Standards (CHEERS) guidelines.**
(PDF)

**S1 Table. Country-level parameter values.** All costs are in 2017 United States Dollars.
(XLSX)

**S2 Table. Regional cost parameters.** All costs are in 2017 United States Dollars.
(PDF)

**S3 Table. Cost per case for healthcare providers and sensitivity analysis excluding productivity losses in children in the baseline global costs, and additional cost results for the *Supervised radical cure* and *Unsupervised radical cure* scenarios.** All costs are in 2017 United States Dollars.
(XLSX)

**S4 Table. Age-stratified case and cost results for the baseline global costs and *Supervised radical cure* and *Unsupervised radical cure* scenarios.** All costs are in 2017 United States Dollars.
(XLSX)

**S1 File. Equations describing the percent reduction in cases for the radical cure scenarios.**
(PDF)

**S2 File. Global assumptions, data sources, and distributions for the probabilistic sensitivity analysis.**
(PDF)

## Acknowledgments

We thank Nick White, Francois Nosten, Kevin Baird, Lorenz von Seidlein, and Cindy Chu for their advice on some of the epidemiological parameters.

## Author Contributions

**Conceptualization:** Angela Devine, Niamh Meagher, Saber Dini, Ric N. Price, Yoel Lubell.

**Data curation:** Angela Devine, Katherine E. Battle, Rosalind E. Howes, Peter W. Gething.

**Formal analysis:** Angela Devine, Niamh Meagher.

**Funding acquisition:** Peter W. Gething, Julie A. Simpson, Ric N. Price.

**Methodology:** Angela Devine, Niamh Meagher, Julie A. Simpson, Ric N. Price, Yoel Lubell.

**Project administration:** Peter W. Gething, Julie A. Simpson, Ric N. Price.

**Software:** Saber Dini.

**Supervision:** Julie A. Simpson, Ric N. Price, Yoel Lubell.

**Visualization:** Angela Devine, Katherine E. Battle.

**Writing – original draft:** Angela Devine, Ric N. Price, Yoel Lubell.

**Writing – review & editing:** Katherine E. Battle, Niamh Meagher, Rosalind E. Howes, Saber Dini, Peter W. Gething, Julie A. Simpson.

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
