## [Editor Report · Decision Letter 0]

30 Sep 2020

Dear Dr Devine, 

Thank you for submitting your manuscript entitled "Global economic costs due to vivax malaria and the potential impact of its radical cure" for consideration by PLOS Medicine.

Your manuscript has now been evaluated by the PLOS Medicine editorial staff and I am writing to let you know that we would like to send your submission out for external peer review.

Kind regards,

Artur Arikainen,

Associate Editor

PLOS Medicine

---

## [Decision Letter · Decision Letter 1]

18 Nov 2020

Dear Dr. Devine,

Thank you very much for submitting your manuscript "Global economic costs due to vivax malaria and the potential impact of its radical cure" (PMEDICINE-D-20-04743R1) for consideration at PLOS Medicine. 

[LINK]

In light of these reviews, I am afraid that we will not be able to accept the manuscript for publication in the journal in its current form, but we would like to consider a revised version that addresses the reviewers' and editors' comments. Obviously we cannot make any decision about publication until we have seen the revised manuscript and your response, and we plan to seek re-review by one or more of the reviewers. 

We expect to receive your revised manuscript by Dec 09 2020 11:59PM. Please email us (plosmedicine@plos.org) if you have any questions or concerns.

We look forward to receiving your revised manuscript. 

Sincerely,

Artur Arikainen, 

Associate Editor 

PLOS Medicine

plosmedicine.org

1. Please address the reviewers’ and academic editor’s comments below.

2. Please revise your title according to PLOS Medicine's style. Your title must be nondeclarative and not a question. It should begin with main concept if possible. "Effect of" should be used only if causality can be inferred, i.e., for an RCT. Please place the study design ("A randomized controlled trial," "A retrospective study," "A modelling study," etc.) in the subtitle (ie, after a colon).

3. Abstract:

a. Around lines 49-50: Please include date ranges for your data.

b. Please perhaps include estimates of patients affected by vivax malaria, and a break-down by world regions.

c. Please rename the “Interpretation” section to “Conclusions”. Please begin this section with “In our modelling study, we found that…”

4. Author Summary: Please briefly explain “vivax malaria” to a lay reader.

5. Please remove spaces from within citation callouts, eg: “…and indirect attributable mortality [4,5] and ongoing…”

6. Methods: Around lines 155-169: Please include date ranges for your data.

7. Line 388: Please clarify here: “To our knowledge, this paper collates for the first time…”

8. Please remove this text from reference 44: “coordinating a workshop in Port Moresby to provide advice on mathematical modelling to the Papua New Guinea National Malaria Control Programme. M.T.W., P.W. and A.G. declare that they have no other competing interests. All remaining authors declare no competing interests.”

9. Please ensure that the study is reported according to the CHEERS guidelines, and include the completed checklist as Supporting Information. When completing the checklist, please use section and paragraph numbers, rather than page numbers. Please mark itens not applicable as “n/a”. Please add the following statement, or similar, to the Methods: "This study is reported as per the Consolidated Health Economic Evaluation Reporting Standards (CHEERS) guideline (S1 Checklist)."

https://www.equator-network.org/wp-content/uploads/2013/04/Revised-CHEERS-Checklist-Oct13.pdf

Academic editor feedback:

1. I agree with the first reviewer that GDP/capita/day probably underestimates the value of clinical providers' time. More important, though, is that it may also incorrectly value households' time, which makes up a fair share of the cost savings identified. From a colleague's paper about the cost of childhood malaria in Kenya: "Per capita income underestimates adult income because children are included in the calculation. However, per capita income overestimates average income because the distribution of income in Kenya is very skewed: 20% of the country's total population is estimated to earn 50% of national income." (Source: PMID 17196105). I would ask the authors to use a more thoughtful estimate of the value of time and to allow it to vary between providers and patients. The paper I cite here has some suggestions for how to do that. Even just providing examples of how this would affect results for a few of the highest burden countries would help.

2. The issue of "cost to whom," while mentioned, is not addressed sufficiently in the discussion. Treating vivax malaria substantially increases costs to providers. All the savings are to households, and a large share of these savings are not cash in hand (i.e. they are indirect benefits of not losing productive time). As the authors note, this makes the intervention look like a bad deal to governments. It also masks the value to households, which might appreciate the saved time but are unlikely to count it as an economic benefit unless there is an immediate income effect. A stronger discussion of how to frame interventions that impose an immediate financial cost on the provider to generate future benefits to patients would be welcome.

---

Comments from the reviewers:

Reviewer #1: Malaria eradication remains a global challenge despite numerous programs and initiatives to accomplish it, and this paper contributes positively by suggesting a fresh angle of attack to this problem. I also agree with the authors that P. vivax has probably received too little attention, and see this manuscript as a potentially important contribution. I also think the main conclusion is plausible that provision of a safe and radical cure is possible, and that it indeed will require and increased investment for national malaria control programs. However, I have some comments to the methods which in consequence means that I think this paper underestimates both the required investments and perhaps overestimate potential benefits. I will present the comments in an order which I perceive as decreasing importance:

 1. My most substantial comment is about the costing of health providers, which are important since they represent the actual investments national health services need to make. The authors have chosen a societal perspective for cost estimation, which is commendable. They include direct and indirect household cost, as well as direct provider costs. The challenge, as I see it, is that the latter appears to be based on a rather narrow identification of provider cost items and therefore probably is underestimated. Provider costs are sometimes limited to facility level, which can be meaningful when comparing health interventions at facility level. The scope of this paper is however to discuss national scale-up of (i) screening and (ii) subsequent treatment. This will require national level efforts to build competence, set up routines for the different procedures and provide and monitor treatment. For example, training is required of the different cadres of laboratory and other health workers to facilitate these interventions. In order to provide repeated supervision of treatments, a system with village health workers need to be operational, sensitised and trained, and their availability probably varies substantially between the included countries. In sum, it appears that none of these systemic costs of scaling up these interventions have been evaluated. This seems apparent in Table 2, where the total provider costs for low-incidence countries such as Belize, Bhutan, China, El Salvador, Iran and Timor-Leste accumulate to between 50 and 200 USD. I recommend that the range of provider costs is re-evaluated in order to reflect more realistic measures for investments of the public health systems.

2. Also pertaining to provider cost is the assumption (line 229) that health workers time is valued using GDP per capita per day. While I think this is a reasonable assumption for national level valuation of productive time, I think this is an underestimate when applied to health workers. Many of the included countries have large informal sectors, such as subsistence farming and petty trade, with generally very low levels of income per capita that bias the national GDP estimates per capita downward. Health workers on the other hand belong to the formal sector, where the income levels are generally substantially higher.

3. Figure 1 is a flow chart illustrating the structure of procedures and costs. Early in the model, eligibility is considered if "patient is not pregnant, lactating or under the age of 1 year". This state may lead to G6PD testing and subsequent treatment. It is however confusing that exactly the same eligibility criteria are repeated further down in the procedures. Why this replication?

4. A different but related confusion is in lines 248-250, where it is explained that the cost of testing was applied to the patient population eligible to receive primaquine. I would expect that the cost of testing applied to all that were tested, not just those with an eligible test result.

5. In line 292, the administration of an anti-relapse dose is explained. This would become clearer if this outcome and procedure was described in the introduction.

6. The paper considers human transmission of P. vivax, but does not mention bovine transmission. How could the presence of bovine transmission influence on the results and conclusion?

7. Line 173 states "RDT or microscopy", while line 175 states "RDT and microscopy". Where both or only one of the procedures performed? This has relevance for costing.

8. Line 189: Please expand acronym FST on its first (and only?) appearance.

9. The paper use US$ exchange rates from 2016. This is fine, but it would have been more eloquent (and more update) to use 2017 rates, since this is also the base year for incidence estimates.

Producing such a paper based on macro-level evidence is challenging, and will always entail many trad-offs between availability and quality of data. I believe this is something we must accept and that research such as this has merit and is of potential importance for policy formulation. The value of such research, in my opinion, lies primarily in its ability to provide a birds perspective of important public health issues rather than representing exact information at local levels. With some methodological improvements, I think this paper is publishable.

Reviewer #2: This interesting manuscript describes the annual global cost of Plasmodium vivax malaria and estimates the cost savings associated with G6PD screening and radical cure assuming perfect and 30% adherence to treatment. Overall, it is an important topic that represents a contribution to the evidence based, but the analysis will be strengthened by addressing the comments noted below.

Adherence scenarios: 

30% adherence is considerably lower than the adherence documented for malaria treatment in most settings. As such, stronger justification should be provided for the selection of that level or a more reasonable alternative scenario should be described. 

Also- the abstract states 30% adherence, but the methods section (line 231 and 241) describes 40% adherence. Please align and clarify what was used. As noted above, a 40% figure seems more appropriate.

Cost time horizon:

The analysis considers the one year cost savings associated with the intervention. Given that the goal of mass screening and administration of radical cure throughout endemic countries, it is reasonable to assume that subsequent years would have further reductions in vivax transmission and, in turn, further savings. This should be better described within the limitations section. Additionally, interventions to improve care-seeking will also improve the impact of the proposed intervention.

Supplementary Table 1- Costs for primaquine in each country should be added alongside the cost of RDT. Please confirm that these costs are currently included in the scenarios as it would be an important additional cost of introduction of radical cure.

Minor comments:

Line 220- I am assuming that the analysis assumes only screening patients who are symptomatic (or positive?) for vivax malaria. This should be explicitly stated. 

Line 228- what was the rationale / evidence for using ¼ of a healthcare worker's time per a case? This seems quite high.

Figure 2- the blues used to denote the most expensive and second most expensive categories are very similar. I suggest changing to make it easier to differentiate India from the other countries.

Table 2- it would be helpful to include an additional column with average cost per a case for each country to provide more evidence about the varied costs by country referenced in line 307.

Line 466 Provision of access to the online application is very useful and an important contribution. 

Reviewer #3: 

This study is concerned with an interesting subject, the cost of vivax malaria. While the general question is important and the methodology is sound, there is a number of issues which require clarification and amendment. Specific comments are given below. 

A major choice in this exercise relates to the selection of the cure method. The authors focus exclusively on primaquine and appear to downgrade the option of treating patients with tafenoquine, which is only briefly mentioned throughout the paper, including a note in the discussion. This is a central/strategic option since using tafenoquine instead would alleviate the need for adherence in practice and the corresponding assumptions in this modelling study. Hence, if the authors will not add the options of treating patients with tafenoquine, a more extensive discussion is in order, with the relative pros and cons of the two treatments, including the option of meticulous (e.g. repeated) testing for G6PD deficiency combined with the use of tafenoquine.

The authors describe their approach reasonably well. However, from a cost-effectiveness standpoint the results are somewhat mixed in that the cost is split, preventing from the reporting of traditional measures like per country ICERs and Incremental Net benefit. Can the authors possibly report such measures, perhaps accounting for the type of country-specific healthcare systems?

The authors chose not to use an epidemic model for estimating the benefit of the proposed approach, therefore underestimating the benefit of indirect protection. I understand that using a full epidemic model may be out of the scope of the present paper, but wonder if the authors could use a multiplier for incorporating the indirect effect as a sensitivity analysis, thus giving a rough order of the complete benefit of the proposed measures, instead of simply calling it a conservative approach.

It would be beneficial to add a table with the critical inputs, including the assumed sensitivity and specificity for each test, giving a sense of their importance in the output of the model. 

Please add a short discussion point regarding the completeness of capturing all the vivax cases and how the treatment-seeking reports may vary by country.

The data in table 1 are slightly puzzling, having the eastern Mediterranean region (EMRO) as the one with the second highest burden. From inspecting the countries of table 2 (and table 4) it is unclear which of those are located in EMRO, is that a typo?

---

[LINK]

---

## [Decision Letter · Decision Letter 2]

25 Mar 2021

Dear Dr. Devine,

Thank you very much for re-submitting your manuscript "Global economic costs due to vivax malaria and the potential impact of its radical cure: A modelling study" (PMEDICINE-D-20-04743R2) for review by PLOS Medicine.

I have discussed the paper with my colleagues and the academic editor and it was also seen again by three reviewers. I am pleased to say that provided the remaining editorial and production issues are dealt with we are planning to accept the paper for publication in the journal.

[LINK]

We expect to receive your revised manuscript by March 29, 2021. Please email us (plosmedicine@plos.org) if you have any questions or concerns.

We look forward to receiving the revised manuscript by Apr 01 2021 11:59PM.   

Sincerely,

Beryne Odeny

Associate Editor 

PLOS Medicine

plosmedicine.org

Requests from Editors:

- Please use the "Vancouver" style to reformat reference #10 and include a link to the online source. See our website for other reference guidelines https://journals.plos.org/plosmedicine/s/submission-guidelines#loc-references

- Please update reference #3

Comments from Reviewers:

Reviewer #2: No additional comments

Reviewer #4: This manuscript provides estimates of the global costs associated with treating P. vivax malaria, and the potential benefit of expanding access to radical cure with primaquine. This manuscript had already been on a journey by the time it landed on my desk, having been read by three other reviewers, with the authors consequently undertaking substantial revisions. Having read the manuscript, the reviewers' comments, and the authors' response, I have found that the majority of the concerns that I would have had have already been addressed by the other reviewers - I thought they did a very thorough job. Nonetheless, I do have some minor comments.

On a more general note, an analysis such as this is undeniably hard, and requires an enormous number of assumptions. Some of these assumptions may be inappropriate or downright wrong - such is the case with all model-based analyses. However, I do find the authors major assumptions to be reasonable and in line with current best practices for this sort of global health analysis. Whether current practices in global health analytics are up to scratch is a different debate. I would probably have done some things differently (e.g. demography, treatment effectiveness, adherence, etc.) but I don't believe any of these would substantially alter the work's findings. At the previous reviewers' request, the authors have included sensitivity analyses, and I find these to be satisfactory. It is tempting to ask for another 100 sensitivity analyses, but I don't think this would bring any additional value. 

In summary, I think this is a solid analysis whose qualities, and flaws, are in line with current best practices in large global health analytics. 

Minor comments

* The online model is a very nice tool.

* Please take care when referring to confidence versus credible intervals. There is a bit of inconsistency.

* I think there's something wrong in Table 1. There may have been a mix up between the first and second iterations of the manuscript (bottom two rows). Please take a very careful look at this.

* I found Table 3 quite unintuitive and it took me a while to work out what was going on. I preferred the format in the first iteration which broke the analyses into several columns.

* The estimates in Table 4 of the percent reduction from baseline are critical, with a lot of variation between countries. Could the authors provide some additional explanation on how these estimates were obtained.

Reviewer #5: This paper describes the total annual global cost of Plasmodium vivax malaria and estimates the cost savings associated with G6PD testing and radical cure assuming perfect and 30% adherence to treatment. This is an important topic and the total cost of Plasmodium vivax is under-studied. Though this could be considered as an additional cost-benefit model with little local implications, I fully agree with R1 that "producing such a paper based on macro-level evidence is challenging, and will always entail many trade-offs between availability and quality of data. I believe this is something we must accept and that research such as this has merit and is of potential importance for policy formulation. The value of such research, in my opinion, lies primarily in its ability to provide a birds perspective of important public health issues rather than representing exact information at local levels. With some methodological improvements". I thus also think that this paper is publishable.

The authors have also taken a careful attention to previous comments made by 3 knowledgeable reviewers and I think their answers are satisfactory. I liked the fact that they consider heterogeneity in their estimates (notably by age) and that they provide an online tool (with their assumptions).

My minor concerns related to this type of exercises are: 

1/ The short-term horizon of the cost savings (one year) associated with the intervention or the full range of costs in the absence of intervention. Long-term versus short-term effects could be discussed more in the paper. In the absence of intervention, the long term effect may not be linear and could intensify or vanish over time. 

2/ The range of costs considered in this study. Many economic studies have shown that vivax malaria control may have a broader range of economic consequences than the one analyzed in a cost-of-illness perspective. The authors do not make reference to such estimates and economic literature. The estimated costs thus certainly represent a low bound in this perspective. See e.g. and amongst other: 

- Bleakley, H. (2010). `Malaria eradication in the Americas: A retrospective analysis of childhood exposure', American Economic Journal: Applied Economics, pp. 1{45.

- Burlando, A. (2015). `The Disease Environment, Schooling, and Development Outcomes: Evidence from Ethiopia', The Journal of Development Studies, vol. 51(12), pp. 1563{1584,ISSN 0022-0388, doi:10.1080/00220388.2015.1087512.

- Lucas, A.M. (2010). `Malaria eradication and educational attainment: evidence from Paraguay and Sri Lanka', American Economic Journal: Applied Economics, vol. 2(2), pp. 46{71.

- Lucas, A.M. (2013). `The impact of malaria eradication on fertility', Economic Development and Cultural Change, vol. 61(3), pp. 607{631.

3/ The potential effects of interventions across areas with different institutions, different risks, different health systems (put differently, externalities or general equilibrium effects) that are only partially addressed.

These comments apply more generally to similar costs of illness studies, and model-based cost-benefits or cost-effectiveness analyses of this kind. It does not mean that they are not useful as mentioned above if one takes them with caution. I do think the discussion is sufficiently cautious here to avoid this caveat.

[LINK]

---

## [Editor Report · Decision Letter 3]

7 Apr 2021

Dear Dr Devine, 

On behalf of my colleagues and the Academic Editor, Dr. Sydney Rosen, I am pleased to inform you that we have agreed to publish your manuscript "Global economic costs due to vivax malaria and the potential impact of its radical cure: A modelling study" (PMEDICINE-D-20-04743R3) in PLOS Medicine.

PRESS

Sincerely, 

Beryne Odeny 

Associate Editor 

PLOS Medicine